# Inference in Probabilistic Answer Set Programs with Imprecise Probabilities via Optimization

**Damiano Azzolini**[1]                    **Fabrizio Riguzzi**[2]

[1]Department of Environmental and Prevention Sciences, University of Ferrara , Ferrara, Italy
[2]Department of Mathematics and Computer Science, University of Ferrara , Ferrara, Italy

## Abstract

Probabilistic answer set programming has recently been extended to manage imprecise probabilities by means of credal probabilistic facts and credal annotated disjunctions. This increases the expressivity of the language but, at the same time, the cost of inference. In this paper, we cast inference in probabilistic answer set programs with credal probabilistic facts and credal annotated disjunctions as a constrained nonlinear optimization problem where the function to optimize is obtained via knowledge compilation. Empirical results on different datasets with multiple configurations show the effectiveness of our approach.

## 1 INTRODUCTION

Uncertainty pervades every aspect of every day's life. Representing these situations with an expressive yet comprehensible language is crucial to understand them. Logic languages, such as Prolog [Sterling and Shapiro, 1994] and Answer Set Programming (ASP) [Brewka et al., 2011] are considered interpretable by design, since users can often encode the domain of interest with few lines of codes. In particular, ASP can compactly describe combinatorial problems. However, these languages can only represent certain data. Several semantics have been proposed to extend logic languages with constructs to represent uncertain data. One of the first was the distribution semantics (DS) [Sato, 1995] that gave the birth to the field of Probabilistic Logic Programming (PLP) and to languages such as ProbLog [De Raedt et al., 2007] and Logic Programs with Annotated Disjunctions (LPADS) [Vennekens et al., 2004]. The credal semantics (CS) [Cozman and Mauá, 2020] applies the ideas of the DS to ASP, obtaining Probabilistic Answer Set Programming (PASP).

Inference in PLP and PASP can be performed via knowl-edge compilation [Darwiche and Marquis, 2002], where the program is converted in an alternative form which allows inference in (possibly) faster way. Recently, Mauá and Cozman [2023] extended PASP with imprecise probabilities, allowing the representation of any credal network with finitely generated credal sets, and propose a solver called dpasp based on vertex enumeration.

In this paper, we propose to solve the inference task in probabilistic answer set programs with imprecise probabilities via optimization, by extracting an equation from the result of knowledge compilation and optimizing it subject to some constraints. Empirical results on 4 different datasets with multiple configurations show that this approach is significantly faster than enumeration adopted in Mauá and Cozman [2023]. Furthermore, our solver is also able to manage annotated disjunctions with imprecise probabilities, which dpasp cannot manage.

The paper is structured as follows: Section 2 introduces background concepts involving PLP and PASP. In Section 3 we discuss how to cast inference in probabilistic answer set programs with imprecise probabilities as an optimization problem. Section 4 presents the experiments conducted to assess the performance of the developed solver. Related works are surveyed in Section 5 and Section 6 concludes the paper.

## 2 BACKGROUND

ProbLog [De Raedt et al., 2007] allows probabilistic facts of the form

$$\Pi :: a$$

where $a$ is a ground atom and $\Pi \in [0, 1]$ is its probability, with the meaning that $a$ is true (resp. false), with probability $\Pi$ (resp. $1-\Pi$). The distribution semantics (DS) [Sato, 1995] is based on the concept of *world*, a normal logic program obtained by including or not each probabilistic fact. If there are $l$ probabilistic facts, then the number of worlds is $2^l$. The

probability of a world $w$ is computed as

$$P(w) = \prod_{a_i \in w} \Pi_i \cdot \prod_{a_i \notin w} (1 - \Pi_i) \qquad (1)$$

where the $\Pi_i :: a_i$ are the probabilistic facts. The probability of an atom $q$, called *query*, is computed as the sum of the probabilities of the worlds where $q$ is true. That is,

$$P(q) = \sum_{w \models q} P(w) \qquad (2)$$

To clarify, consider Example 1.

**Example 1** *The following probabilistic logic program has two probabilistic facts: $a$ with probability 0.3 and $b$ with probability 0.4.*

```
0.3::a.
0.4::b.
q:- a.
q:- b.
```

*It has four worlds: $w_1$ where both $a$ and $b$ are false, with $P(w_1) = (1 - 0.3) \cdot (1 - 0.4) = 0.42$; $w_2$ where $a$ is false and $b$ is true, with $P(w_2) = (1 - 0.3) \cdot 0.4 = 0.28$; $w_3$ where $a$ is true and $b$ is false, with $P(w_3) = 0.3 \cdot (1 - 0.4) = 0.18$; and $w_4$ where both $a$ and $b$ are true, with $P(w_4) = 0.3 \cdot 0.4 = 0.12$. The probability of the query $q$ is given by $P(q) = P(w_2) + P(w_3) + P(w_4) = 0.28 + 0.18 + 0.12 = 0.58$, since $q$ is true in all the worlds except for $w_1$.*

Annotated disjunctions were introduced by Vennekens et al. [2004] with the syntax $h_1 : \Pi_1; \ldots; h_m : \Pi_m :- b_1, \ldots, b_n$, where $\sum_i \Pi_i = 1$. The meaning is that, when the conjunction of the literals $b_j$ in the body is true, one of the head atoms $h_i$ is true with the corresponding probability $\Pi_i$. We consider the notation

$$\Pi_1 :: h_1; \ldots; \Pi_m :: h_m :- b_1, \ldots, b_n$$

for uniformity with ProbLog. Annotated disjunctions can be converted into probabilistic facts [De Raedt et al., 2008] as follows: for each annotated disjunction $\Pi_1 :: h_1; \ldots; \Pi_m :: h_m :- B$ (with $B$ the body) with $m$ heads we add $m - 1$ probabilistic facts and $m$ rules:

$$
\begin{aligned}
&\pi_1 :: f_1. \\
&\ldots \\
&\pi_{m-1} :: f_{m-1}. \\
&h_1 :- B, f_1. \qquad\qquad\qquad (3)\\
&h_2 :- B, not\ f_1, f_2. \\
&\ldots \\
&h_m :- B, not\ f_1, \ldots, not\ f_{m-1}.
\end{aligned}
$$

where $\pi_1 = \Pi_1$ and $\pi_i = \Pi_i / \sum_{j=1}^{i-1}(1 - \pi_i)$ for $i > 0$. For example, the probabilistic facts obtained by converting

the annotated disjunction $0.2 :: a; 0.3 :: b; 0.5 :: c$ have probability 0.2 and $0.3/(1 - 0.2) = 0.375$.

The credal semantics (CS) assigns a meaning to probabilistic answer set programs [Cozman and Mauá, 2020], i.e., answer set programs [Brewka et al., 2011] extended with ProbLog probabilistic facts. In this setting, a world may have zero or more stable models (also called answer sets). A stable model is a minimal model under set inclusion of the reduct of an answer set program $P$, where the reduct of $P$ w.r.t. an interpretation $I$ is obtained by removing from $P$ all the rules that have the body false in $I$. An interpretation is called model if it satisfies all the groundings of the rules of $P$. The CS requires that every world has at least one answer set. The probability of a query $q$ under the CS is described by a probability range, defined by a lower ($\underline{P}(q)$) and an upper ($\overline{P}(q)$) bound. A world $w$ contributes to both the lower and upper bound if the query is true in *every* answer set of $w$. Conversely, a world $w$ contributes to only the upper bound if the query is true in *at least one* answer set of $w$. This can be expressed in formulas as

$$\underline{P}(q) = \sum_{w_i | \forall m \in AS(w_i),\ m \models q} P(w_i) \qquad (4)$$

$$\overline{P}(q) = \sum_{w_i | \exists m \in AS(w_i),\ m \models q} P(w_i) \qquad (5)$$

**Example 2** *The following probabilistic answer set program is a variation of the probabilistic logic program shown in Example 1.*

```
0.3::a.
0.4::b.
q:- a.
q ; r :- b.
```

*Consider, as in Example 1, the query $q$. $w_2$ (where $a$ is false and $b$ is true) has 2 answer sets, $\{q, b\}$ and $\{r, b\}$, and it only contributes to the upper probability for $q$, since $q$ is true only in the first. $w_3$ and $w_4$ have a unique answer set each ($\{a, q\}$ and $\{a, b, q\}$, respectively) where $q$ is true, so they contribute to both lower and upper probability. $w_1$ has a unique answer set with no atoms, so it does not contribute. Overall, $P(q) = [\underline{P}(q), \overline{P}(q)] = [P(w_3) + P(w_4), P(w_2) + P(w_3) + P(w_4)] = [0.3, 0.58]$.*

Inference in probabilistic answer set programs can be performed via Second Level Algebraic Model Counting (2AMC) [Kiesel et al., 2022]. 2AMC is an abstract framework that comprises several well-known problems, such as inference, decision theoretic inference, and MAP inference. The task considers two commutative semirings [Gondran and Minoux, 2008] $R^i = \{D^i, \oplus^i, \otimes^i, n_{\oplus^i}, n_{\otimes^i}\}$ and $R^o = \{D^o, \oplus^o, \otimes^o, n_{\oplus^o}, n_{\otimes^o}\}$, a propositional theory $T$ whose variables are split into two disjoint sets, $V_o$ and $V_i$,

two weight functions $w_i$ and $w_o$, and a transformation function $f$, and requires solving:

$$2AMC(T) = \bigoplus_{I_o \in \mu(V_o)}^o \bigotimes_{a \in I_o}^o w_o(a) \otimes^o$$
$$f(\bigoplus_{I_i \in \varphi(\Pi|I_o)}^i \bigotimes_{b \in I_i}^i w_i(b)) \quad (6)$$

where $\mu(V_o)$ is the set of possible assignments to the variables in $V_o$ and $\varphi(T \mid I_o)$ is the set of possible assignments to the variables in $T$ that satisfy $I_o$. Said otherwise, for every assignment of the variables $V_o$, we need to solve an inner AMC [Kimmig et al., 2017] task on the variables $V_i$. The result of the inner AMC is converted through the transformation function $f$ into a value coherent with the ones of the variables in the outer semiring and we need to solve a second AMC task, this time by considering $V_o$. Kiesel et al. [2022] also introduced a tool called aspmc [Eiter et al., 2021] to solve 2AMC problems, based on a conversion guided by the treewidth of the program into a compact form via knowledge compilation [Darwiche and Marquis, 2002]. The target of the compilation is negation normal form (NNF), a rooted directed acyclic graph where internal nodes are associated with conjunction (AND-nodes) or disjunctions (OR-nodes) and leaves are associated with literals, true, or false. Furthermore, the obtained NNF has the properties of decomposability, determinism, smoothness, and X-first modulo definability. In particular, the last property imposes an hierarchical structure on the appearance of the variables, allowing to consider them according to the specified semiring. This tree can be traversed bottom up to solve the 2AMC task.

Azzolini and Riguzzi [2023] introduced aspcs, a solver built on top of aspmc, to perform inference in probabilistic answer set programs. To do so, they instantiate Equation 6 by considering as inner semiring $\mathcal{R}_{in} = (\mathbb{N}^2, +, \cdot, (0, 0), (1, 1))$ with $w_i$ mapping $not\ q$ (assuming that we are interested in computing the probability of the query $q$) to $(0, 1)$ and all other literals to $(1, 1)$, where operations are performed component-wise. In this inner semiring, we have a fixed world. Practically, $\mathcal{R}_{in}$ counts the number of answer sets where the query is true and the total number of answer sets. The transformation function is $f(n_1, n_2)$ returning a pair $(v_{lp}, v_{up})$ where $v_{lp} = 1$ if $n_1 = n_2$, 0 otherwise, and $v_{up} = 1$ if $n_1 > 0$, 0 otherwise. This function checks the bounds to which the considered world contributes. The outer semiring is an extension of the probability semiring [Kimmig et al., 2017] to two dimensions, i.e., $\mathcal{R}_{out} = ([0, 1]^2, +, \cdot, (0, 0), (1, 1))$ (the operations are still considered component-wise), with $w_o$ associating $(p, p)$ and $(1 - p, 1 - p)$ to $a$ and $not\ a$, respectively, for every probabilistic fact $p :: a$ and $(1, 1)$ to all the remaining literals. In other words, in $\mathcal{R}_{out}$, we multiply the probabilities of the probabilistic facts and sum them to obtain the probability of $q$.

Mauá and Cozman [2023] introduced probabilistic facts

with an imprecise probability, that in this paper we call *credal facts*. These are expressed with the syntax $[\alpha, \beta] :: a$ with the meaning that $a$ (a ground atom) has a probability ranging between $\alpha$ and $\beta$, with $\alpha, \beta \in \mathbb{R}$, $0 \leq \alpha \leq \beta \leq 1$. Mauá and Cozman [2023] also discussed annotated disjunctions with imprecise probabilities, that we call *credal annotated disjunctions*, for uniformity with credal facts. Their syntax is

$$[\alpha_1, \beta_1] :: h_1; \ldots; [\alpha_m, \beta_m] :: h_m :- b_1, \ldots, b_n \quad (7)$$

where each $h_i$ is an atom and each $b_j$ is a literal. The meaning is that, when the conjunction of the literals in the body is true, one of the head atoms is true with the corresponding probability range. To ensure well definedness, the reachability property must be met, i.e., $\alpha_i + \sum_{j \neq i} \beta_j \geq 1$ and $\beta_i + \sum_{j \neq i} \alpha_j \leq 1$ for all $i$.

## 3 INFERENCE WITH CREDAL FACTS AND CREDAL ANNOTATED DISJUNCTIONS

In this section, we first discuss how to cast inference in programs with credal facts only (and possibly probabilistic facts) as an optimization problem. Then, we extend that approach to also manage credal annotated disjunctions.

### 3.1 INFERENCE WITH CREDAL FACTS

Inference in programs with credal facts and without credal annotated disjunctions can be cast as inference in probabilistic answer set programs with only probabilistic facts, by converting (as described by [Mauá and Cozman, 2023]) each credal fact $[\alpha, \beta] :: a$ into an annotated disjunction $\alpha :: a1; 1 - \beta :: a2; \beta - \alpha :: a3$ and three rules $a :- a1$, $na :- a2$, and $a; na :- a3$. To clarify, consider Example 3.

**Example 3** *The following probabilistic answer set program is a variation of one shown in Example 2. Now, a and b are credal facts.*

```
[0.3,0.4]::a.
[0.4,0.9]::b.
q:- a.
q ; r :- b.
```

*This program is converted into*

```
0.3 :: a1 ; 0.6 :: a2 ; 0.1 :: a3.
a :- a1.
na :- a2.
a ; na :- a3.
0.4 :: b1 ; 0.1 :: b2 ; 0.5 :: b3.
b :- b1.
nb :- b2.
```

```
b ; nb :- b3.
q:- a.
q ; r :- b.
```

*The probability of the query q is [0.3, 0.94].*

However, this conversion greatly increases the number of probabilistic facts and rules, possibly slowing down the inference process. We propose an alternative approach based on 2AMC. We consider as inner semiring and transformation function the same of Azzolini and Riguzzi [2023] and described in Section 2. For the outer semiring $\mathcal{R}_{out}$ we propose an extension to two dimensions of the sensitivity semiring [Kimmig et al., 2017]: $\mathcal{R}_{out} = (\mathbb{R}[X], +, -, (0, 0), (1, 1))$, with $w_o$ associating $(p_i, p_i)$ and $(1 - p_i, 1 - p_i)$ to $a_i$ and $not\ a_i$, respectively, for every probabilistic fact $p_i :: a_i$; $(\pi_i, \pi_i)$ and $(1 - \pi_i, 1 - \pi_i)$ to $v_i$ and $not\ v_i$, respectively, for every credal fact $[\alpha, \beta] :: v_i$, where the $\pi_i$ are symbolic variables representing the probability of the credal facts. The remaining literals are assigned $(1, 1)$. In this way, by solving the 2AMC, we obtain two non-linear formulas with at most $k$ variables, where $k$ is the number of credal facts, composed by summations and products only (except for trivial cases). Let us call them $f_{lp}(X)$ and $f_{up}(X)$, where $X = \{\pi_1, \ldots, \pi_k\}$. Their general form is $f_{lp/up}(X) = \sum_w \prod_{a_i \in w} \pi_i \prod_{a_j \notin w} (1 - \pi_j)$. If we replace each variable with a value, we get the lower and upper probability for the query when the facts have the chosen values. Now, the lower probability for $q$ can be obtained by minimizing $f_{lp}(X)$ where the variables $\pi_1, \ldots, \pi_k$ have the range described by the credal fact they represent. That is, if $\pi_1$ is associated to the credal fact $[0.2, 0.6] :: a_i$, $\pi_1$ must be in the range $[0.2, 0.6]$. Similarly, the upper probability for $q$ can be obtained by maximizing $f_{up}(X)$ (or, equivalently, minimizing $-f_{up}(X)$) where the bounds of the variables are the same as before. In other words, we cast inference in programs with credal facts as a nonlinear optimization problem. That is, $\underline{P}(q)$ can be obtained by solving

$$\begin{aligned} minimize \quad & f_{lp}(X) \\ s.t. \quad & \pi_i \in [l_i, u_i], \forall i \in \{1, \ldots, k\} \end{aligned} \quad (8)$$

while $\overline{P}(q)$ by solving

$$\begin{aligned} maximize \quad & f_{up}(X) \\ s.t. \quad & \pi_i \in [l_i, u_i], \forall i \in \{1, \ldots, k\} \end{aligned} \quad (9)$$

In this way, we avoid the explosion of the size of the program due to the introduction of auxiliary annotated disjunctions.

**Example 4** *Consider the program with credal facts shown in Example 3. It is converted into*

```
pa::a.
pb::b.
q:- a.
q ; r :- b.
```

*By traversing the NNF for the query q we extract two equations: $f_{lp}(pa) = pa$ and $f_{up}(pa, pb) = pa - pb \cdot (pa - 1)$, the former for the lower probability and the latter for the upper probability. To compute $\underline{P}(q)$, we need to minimize $f_{lp}(pa)$ with $pa \in [0.3, 0.4]$. Clearly, the minimum value of $f_{lp}(pa)$ is 0.3. Consider now the computation of $\overline{P}(q)$. We need to maximize $f_{up}(pa, pb)$, or equivalently, minimize $-f_{up}(pa, pb)$ with $pa \in [0.3, 0.4]$ and $pb \in [0.4, 0.9]$. In this case, the maximum value of $f_{up}(pa, pb)$ is 0.94. Thus, $[\underline{P}(q), \overline{P}(q)] = [0.3, 0.94]$.*

### 3.2 INFERENCE WITH CREDAL ANNOTATED DISJUNCTIONS

When credal annotated disjunction are present in a program, the conversion between annotated disjunctions and probabilistic facts (see Section 2) does not preserve the equivalence, as discussed in [Mauá and Cozman, 2023]. Still in [Mauá and Cozman, 2023], to perform inference, the authors propose to convert each credal annotated disjunction with $k$ disjunctions in the head into a set of $k$ credal facts with vacuous intervals (i.e., $\alpha = 0$ and $\beta = 1$), a set of deterministic rules, and a set of annotated disjunctions representing all the vertices of the induced credal set. The set of annotated disjunctions (as well as the set of deterministic rules) obtained with this conversion may be exponential in the number of credal facts, making inference very expensive. If we consider the example discussed in [Mauá and Cozman, 2023], the credal annotated disjunction $D = [0.1, 0.3] :: red; [0.2, 0.4] :: green; [0.4, 0.6] :: blue$, is converted into three credal facts with vacuous intervals, 10 deterministic rules, and 6 annotated disjunctions with 3 atom each in the head.

Here we follow another path: we convert each credal annotated disjunction into a set of probabilistic facts and rules, as explained in Section 2. However, we do not compute the probabilities of the obtained probabilistic facts but we leave them symbolic. More precisely, for a credal annotated disjunction of the form of Equation 7, we get the rules of Equation 3. That is, from $D$ discussed few lines above we obtain: $\pi_1 :: f_1, \pi_2 :: f_2, red :- f_1, green :- not\ f_1, f_2$, and $blue :- not\ f_1, not\ f_2$. Note again that the probability of $f_1$ and $f_2$ is kept symbolic. To compute the probability of a query $q$, we extract, as described in Section 3.1, two symbolic equations, one for the lower, $f_{lp}(X)$, and one for the upper probability, $f_{up}(X)$. As before, we treat the inference process as an optimization problem: $\underline{P}(q)$ can be computed by minimizing $f_{lp}(X)$ while $\overline{P}(q)$ by maximizing $f_{up}(X)$. Both problems, however, require imposing an additional set of nonlinear constraints that mimics the probability conversion between annotated disjunctions and probabilistic facts (Section 2). With $n_{ad}$ credal annotated disjunctions of the form of Equation 7, the constraints can be compactly

expressed as

$$minimize \quad f(X)$$
$$s.t. \quad \pi_i^l \cdot \prod_{j<i}(1 - \pi_j^l) - \alpha_i^l \geq 0,$$
$$\beta_i^l - \pi_i^l \cdot \prod_{j<i}(1 - \pi_j^l) \geq 0,$$
$$\forall l \in \{1, \ldots, n_{ad}\}, \ \forall i \in \{1, \ldots, m_l\}$$

assuming $\pi_i^{m_l} = 1$, where $\pi_i^k$ is the probability associated to the $i$-th probabilistic fact related to the $i$-th head of the $k$-th annotated disjunction. Overall, for a credal annotated disjunction with $m$ heads we add $2 \cdot m$ constraints. So, the problem of inference can be cast as two nonlinear optimization problems with nonlinear constraints. For example, with a credal annotated disjunction of the form $[\alpha_1, \beta_1] :: h_1; [\alpha_2, \beta_2] :: h_2; [\alpha_3, \beta_3] :: h_3$ call $\pi_1$ and $\pi_2$ the probabilities associated with the probabilistic facts obtained via the conversion discussed in Section 2, we have $\pi_1 - \alpha_1 \geq 0$, $\beta_1 - \pi_1 \geq 0$, $(1 - \pi_1) \cdot \pi_2 - \alpha_2 \geq 0$, $\beta_2 - (1 - \pi_1) \cdot \pi_2 \geq 0$, $(1 - \pi_1) \cdot (1 - \pi_2) - \alpha_3 \geq 0$, and $\beta_3 - (1 - \pi_1) \cdot (1 - \pi_2) \geq 0$. More concretely, with the credal annotated disjunction $D$, we obtain the set of constraints: $c_1 - 0.1 \geq 0, 0.3 - \pi_1 \geq 0, (1 - \pi_1) \cdot \pi_2 - 0.2 \geq 0$, $0.4 - (1 - \pi_1) \cdot \pi_2 \geq 0, (1 - \pi_1) \cdot (1 - \pi_2) - 0.4 \geq 0$, and $0.6 - (1 - \pi_1) \cdot (1 - \pi_2) \geq 0$. Note that this approach can be straightforwardly extended to support *parameterized* annotated disjunctions, i.e., annotated disjunctions where the probabilities of the heads are specified via a set of constraints, for example $A :: a; B :: b; C :: c :- A < B, B < C, C > 0.1$, where $A$, $B$ and $C$ are the probabilities to be determined. In our framework, this consists in adding the set of constraints specified in the body to the optimization problem.

## 4 EXPERIMENTS

We implemented the proposed algorithm in Python on top of the aspcs solver [Azzolini and Riguzzi, 2023] and leveraged the SciPy library [Virtanen et al., 2020] to solve the optimization problems and SymPy [Meurer et al., 2017] to simplify the equation extracted from the NNF[1]. We tested both the COBYLA [Powell, 1994] and SLSQP [Kraft, 1994] algorithms for nonlinear constrained optimization, without changing the default parameters. We compare it against dpasp [Mauá and Cozman, 2023], which adopts vertex enumeration and only supports credal facts (not credal annotated disjunctions). We run the evaluation on a machine running at 2.40 GHz with an execution time limit of 8 hours and 16 GB of RAM.

We set up two different set of experiments: in a first round, call it $E_1$, we consider four datasets with probabilistic and

[1]Source code and datasets available at: `https://github.com/damianoazzolini/aspmc`.

credal facts and compare aspcs against dpasp; in a second round, call it $E_2$, we consider a variation of the same four datasets but with credal annotated disjunctions instead of credal facts. dpasp does not support these, so we only report the results for aspcs. All the instances for all the datasets have at least one answer set per world, as required by the credal semantics.

Let us start by describing the four datasets for $E_1$, adapted from [Azzolini and Riguzzi, 2023]. The programs for aspcs and dpasp only differ in the negation symbol: $\backslash+$ for the former and *not* for the latter. The following snippets show the aspcs version. The first dataset, *coloring*, encodes a graph coloring scenario, a well-known problem that can be easily modeled in ASP. Here, some nodes can be associated with three distinct colors, namely red, green, and blue, and others have a fixed color. Nodes are connected by credal probabilistic facts $edge/2$ and connected nodes must have different colors. All the instances have the same following rules:

```
red(X) :- node(X), \+ green(X),
  \+ blue(X).
green(X) :- node(X), \+ red(X),
  \+ blue(X).
blue(X) :- node(X), \+ red(X),
  \+ green(X).
e(X,Y) :- edge(X,Y).
e(Y,X) :- edge(Y,X).
:- e(X,Y), red(X), red(Y).
:- e(X,Y), green(X), green(Y).
:- e(X,Y), blue(X), blue(Y).
```

but they differ in the number of nodes and number of edges (this value denotes the size of the instance). Each instance has an additional set of rules $qr :- blue(i), \forall i \in \{1, \ldots, n\}$, where $n$ is the number of nodes. The query is $qr$.

The *smoke* dataset, introduced in Totis et al. [2023], encodes a network of people where some of them are smokers while other smoke only due to the influence of their friends. All the programs have the following rules:

```
asthma_r(X):- smokes(X), asthma_f(X).
asthma(X):- asthma_f(X).
asthma(X):- asthma_r(X).
asthma_and_stress(X):-
  stress(X), stress_f(X), asthma(X).
smokes(X):- infl(Y,X), smokes(Y).
smokes(X):- asthma_and_stress(X),
  \+ no_smokes(X).
no_smokes(X):- asthma_and_stress(X),
  \+ smokes(X).
```

where $asthma\_f/1$, $stress/1$, $stress\_f/1$, and $asthma\_f/1$ are probabilistic with a sharp probabil-

ity value associated (and there is one of them for each people involved) while the $infl/2$ facts are credal probabilistic facts. The instances have an increasing number of people, and thus an increasing number of probabilistic and credal facts. The query is $smokes(1)$.

The dataset *irl* contains a set of instances with three clauses with an increasing number of credal probabilistic facts $a_i$ in the body: $qr :- \wedge_{i<k,i\ even}\, a_i, qr :- \wedge_{i<k,i\ odd}\, a_i, \backslash + nqr$, and $nqr :- \wedge_{i<k,i\ odd}\, a_i, \backslash + qr$, where $k$ denotes the size of the instance. The instance of size 5 is

```
qr:- a0, a2, a4.
qr:- a1, a3, \+ nqr.
nqr:- a1, a3, \+ qr.
```

where the $a_i$, $i \in \{0, \dots, 4\}$, are credal facts. The query is $qr$.

Each instance $k$ of the fourth dataset, *irn*, contains a set of $qr :- a_i$ rules $\forall i \in \{0, \dots, k-1\}$ even, $qr :- a_i, \backslash + nqr \;\forall i \in \{0, \dots, k-1\}$ odd, and $nqr :- a_i, \backslash + qr \;\forall i \in \{0, \dots, k-1\}$ odd. For example, the instance of size 5 is

```
qr:- a0. qr:- a2. qr:- a4.
qr :- a1, \+ nqr. nqr :- a1, \+ qr.
qr :- a3, \+ nqr. nqr :- a3, \+ qr.
```

The query is $qr$.

For all the datasets we consider two variations, called *loose* and *strict*: for the former, all the credal facts are associated with the probability range [0.05,0.95]; for the latter, the range is [0.45,0.55]. This is because we want to evaluate the optimization process with different types of constraints.

Since the equations extracted from the NNF usually involve many products and summations and this may possibly slow down the optimization process, in a preliminary test on the *coloring* dataset we compared the execution time of aspcs with and without simplifying them by using Sympy. In particular, we are interested in assessing whether the extra execution time spent to simplify the equations effectively speeds up the overall execution time. Table 1 shows the total execution time of running aspcs with the COBYLA algorithm on the *coloring* dataset in both strict and loose version with and without simplification: for the largest solvable instance (size 16) the execution time with simplification is 6 times less than the one without. This proves that the simplification is a crucial step in the pipeline. Note that the simplification is performed twice, once for the lower and once for the upper probability. So, in the remaining experiments, we always consider aspcs with simplification.

To investigate even further the impact of the different steps involved, Table 2 reports the NNF computation time, equations simplification time, and optimizations time for the COBYLA algorithm on the loose configuration of the *coloring* dataset. For bigger instances, the simplification time

Table 1: Impact of the simplification process on the probability computation with the COBYLA algorithm on the *coloring* dataset. The columns represent, respectively, the size of the instance ($size$), the execution time (seconds) for the strict version with simplification ($s.\ w.$), the execution time (seconds) for the strict version without simplification ($s.\ wo.$), the execution time (seconds) for the loose version with simplification ($l.\ w.$), and the execution time (seconds) for the strict version without simplification ($l.\ wo.$).

| $size$ | $s.\ w.$ | $s.\ wo.$ | $l.\ w.$ | $l.\ wo.$ |
|---|---|---|---|---|
| 8 | 4.428 | 5.727 | 5.39 | 6.612 |
| 9 | 5.094 | 6.981 | 5.071 | 7.436 |
| 10 | 7.536 | 13.284 | 8.014 | 14.505 |
| 11 | 13.406 | 26.228 | 12.836 | 27.313 |
| 12 | 33.116 | 47.158 | 32.947 | 47.298 |
| 13 | 46.987 | 103.322 | 46.778 | 126.434 |
| 14 | 46.995 | 164.814 | 46.747 | 172.029 |
| 15 | 84.338 | 308.938 | 84.876 | 362.038 |
| 16 | 104.057 | 627.793 | 104.631 | 675.879 |

Table 2: Impact in seconds and percentage on the total execution time of NNF construction ($NNF$), simplification ($simpl$), and optimization ($opt$) on the probability computation with the COBYLA algorithm on the loose configuration of the *coloring* dataset. The percentages do not sum to 100 due to other internal operations.

| $size$ | $NNF(s)$ | $simpl.(s)$ | $opt.(s)$ |
|---|---|---|---|
| 8 | 2.192 (40.66%) | 1.449 (26.88%) | 0.058 (1.07%) |
| 9 | 3.250 (64.08%) | 0.546 (10.76%) | 0.064 (1.26%) |
| 10 | 4.399 (54.89%) | 2.008 (25.05%) | 0.134 (1.67%) |
| 11 | 9.252 (72.07%) | 2.257 (17.58%) | 0.095 (0.74%) |
| 12 | 29.68 (90.08%) | 1.859 (5.642%) | 0.076 (0.23%) |
| 13 | 35.04 (74.90%) | 10.27 (21.95%) | 0.162 (0.34%) |
| 14 | 35.88 (76.75%) | 9.303 (19.90%) | 0.212 (0.45%) |
| 15 | 43.17 (50.86%) | 39.97 (47.09%) | 0.226 (0.26%) |
| 16 | 46.31 (44.26%) | 56.56 (54.05%) | 0.321 (0.30%) |

requires more than half of the total execution time (again, the reported value is the sum of two simplifications, one for the equation for the lower probability and one for the equation for the upper probability). However, this makes solving the optimizations (also here the time represents optimization applied on both equations) very quick (less than a second), compared to the hundreds of seconds required in the case the equation is not simplified (see Table 1). We report these detailed values only for the *coloring* dataset, due to space restrictions. However, these considerations apply to all the datasets.

We are now ready to discuss the results for $E_1$. Figure 1 shows the results on the *coloring* dataset for all the configurations. aspcs is faster than dpasp and COBYLA and SLSQP for both loose and strict configurations have similar execution times. The instance of size 16 was the largest

solvable one. For dpasp, as expected, the loose and strict configurations require the same time to query. The figure also reports the execution times for the converted instances, where credal facts were converted into annotated disjunctions. Also in this case aspcs outperforms dpasp, and, for both, inference in the program with credal facts is substantially faster than inference in the converted program. This is probably due to the large number of probabilities introduced by the annotated disjunctions. In this case, the largest solvable instance had size 13 for both. For dpasp we only plot up to size 11 since 12 and 13 required respectively 3340 and 25083 seconds and would have made all the other curves in the figure unreadable.

Figure 2 shows the results on the *smoke* dataset. As for the *coloring* dataset, aspcs is faster than dpasp. For dpasp, the loose configuration of instance 6 is approximately 500 seconds faster than the strict. aspcs requires approximately 60 seconds to run the converted version. dpasp was also able to solve it in 26861 seconds (that we do not plot). Interestingly, the converted version for the strict SLSQP configuration is slightly faster than the non-converted one: this may be due to knowledge compilation that can find a good NNF structure for the program.

The results for *irl* are shown in Figure 3. Here, aspcs can solve the instance of size 20 with both COBYLA and SLSQP in both configurations in less than 10 seconds, while dpasp requires more than 1500 seconds. For the converted version, the largest solvable instance size was 15: also in this case aspcs shows better performances than dpasp. This proves that knowledge compilation can find a good representation of the program, since, for example, the equation for the upper probability is composed by the product of all the probabilities of the credal facts.

Lastly, Figure 4 shows the results for *irn*: it confirms the trend of the three previous datasets, where aspcs is faster than dpasp and the converted version is slower than the version with credal probabilistic facts.

For $E_2$, we modify the four datasets to include credal annotated disjunctions. For the *coloring* dataset, we add a credal annotated disjunction $[0.1, 0.3] :: c0(X); [0.2, 0.4] :: c1(X); [0.4, 0.6] :: c2(X)$ for every node $X$ and modify the constraint $:- e(X, Y), blue(X), blue(Y)$ into $:- e(X, Y), blue(X), blue(Y), c2(X), c2(Y)$. Furthermore, edges are now considered probabilistic facts. The remaining rules are the same. For *smoke*, as for *coloring*, we add a credal annotated disjunction $[0.1, 0.3] :: c0(X); [0.2, 0.4] :: c1(X); [0.4, 0.6] :: c2(X)$ and replace the rule $smokes(X) :- infl(Y, X), smokes(Y)$ with $smokes(X) :- infl(Y, X), smokes(Y), c0(X), c1(Y)$ and consider the *infl* facts probabilistic with a sharp probability value. Lastly, for *irl* and *irn* we replace each credal probabilistic fact $a_i$ with: i) $[0.1, 0.3] :: a_i; [0.2, 0.4] :: a_{i1}(X); [0.4, 0.6] :: a_{i2}(X)$ if $i\%3 = 0$; ii) $[0.1, 0.3] ::$

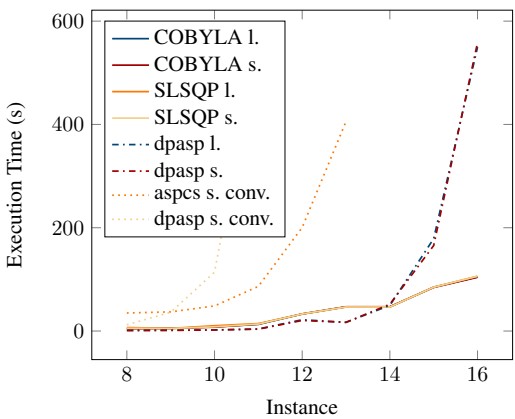

Figure 1: Execution times for the *coloring* datasets for aspcs (solid lines) and dpasp (dash-dotted lines). Dotted lines are the results obtained in programs where the credal facts are converted into annotated disjunctions.

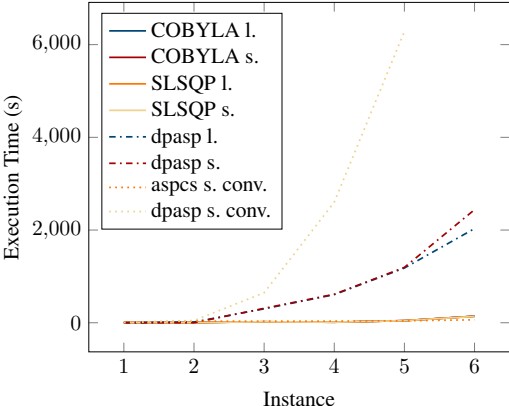

Figure 2: Execution times for the *smoke* datasets for aspcs (solid lines) and dpasp (dash-dotted lines). Dotted lines represent the results obtained in the programs where the credal facts were converted into annotated disjunctions.

$a_{i1}; [0.2, 0.4] :: a_i(X); [0.4, 0.6] :: a_{i2}(X)$ if $i\%3 = 1$; or ii) $[0.1, 0.3] :: a_{i1}; [0.2, 0.4] :: a_{i2}(X); [0.4, 0.6] :: a_i(X)$ if $i\%3 = 2$. The rules are the same as for $E_1$.

Table 3 shows the results. The algorithm can solve fewer instances than the version with only credal facts: this is due to the increasing number of probabilistic facts that need to be considered, whose number is indicated in the fourth column. Furthermore, the COBYLA algorithm is always, except for one case, faster than SLSQP.

## 5 RELATED WORK

Apart from the credal semantics, there exist various alternatives to represent uncertainty with answer set programs, such as $LP^{MLN}$ [Lee and Wang, 2016], P-log [Baral et al., 2009], and diff SAT [Nickles, 2018], but imprecise probabil-

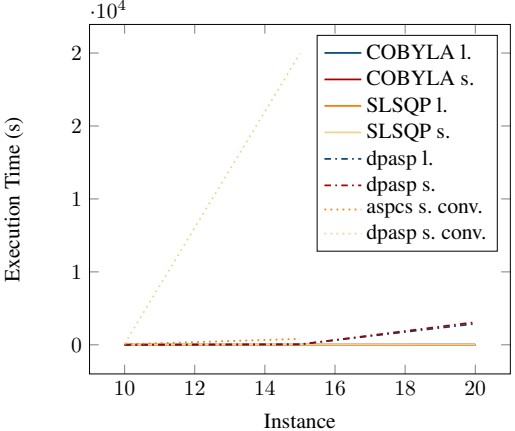

Figure 3: Execution times for the *irl* datasets for aspcs (solid lines) and dpasp (dash-dotted lines). Dotted lines represent the results obtained in the programs where the credal facts were converted into probabilistic facts.

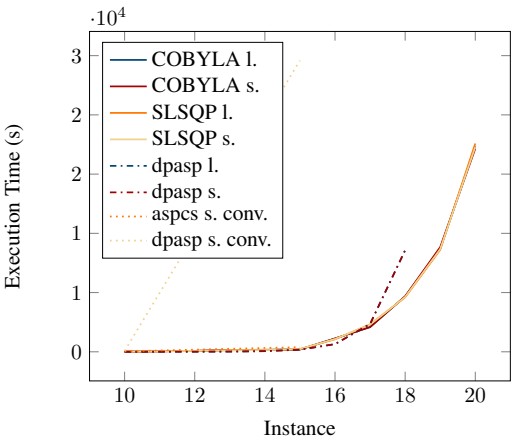

Figure 4: Execution times for the *irn* datasets for aspcs (solid lines) and dpasp (dash-dotted lines). Dotted lines represent the results obtained in the programs where the credal facts were converted into probabilistic facts.

ities have not been proposed for these. Also the approaches of Totis et al. [2023] and Rocha and Cozman [2022] are probabilistic extensions of ASP, both targeting argumentation, but they still do not consider probabilistic fact with imprecise probabilities. As previously discussed, Mauá and Cozman [2023] introduced credal probabilistic facts and credal annotated disjunctions and developed a solver based on vertex enumeration, that, however, does not yet support credal annotated disjunctions. Azzolini and Riguzzi [2021] integrated constrained optimization within probabilistic logical inference, but they considered PLP, not PASP, and a different optimization framework. Constraints are a standard component of ASP solvers, but they are limited to integers (i.e., floating points are usually not supported, even if some alternatives are under development [Pacenza and Zangari, 2023]) and probabilities are not managed. Some works on

Table 3: Execution times in seconds for the experiments with credal annotated disjunctions. The last two columns contain the number of probabilistic facts and the number of rules in each instance, respectively. Lowest execution times are in bold.

| $size$ | $cobyla\ (s)$ | $slsqp\ (s)$ | $\#pf$ | $\#rules$ |
|---|---|---|---|---|
| *coloring* | | | | |
| 8 | **45.135** | 55.151 | 20 | 14 |
| 9 | **104.984** | 105.839 | 23 | 15 |
| 10 | **503.691** | 504.313 | 24 | 15 |
| *irl* | | | | |
| 5 | **5.084** | 12.438 | 10 | 18 |
| 10 | **24.915** | 56.726 | 20 | 33 |
| 15 | **755.598** | 877.51 | 30 | 48 |
| 20 | 12132.332 | **11956.459** | 40 | 63 |
| *irn* | | | | |
| 10 | **264.298** | 284.184 | 20 | 45 |
| 11 | **1339.725** | 2184.979 | 22 | 49 |
| 12 | **4525.672** | 4837.060 | 24 | 54 |
| 13 | **4752.085** | 5994.453 | 26 | 58 |
| 14 | **14627.982** | 15384.466 | 28 | 63 |
| *smoke* | | | | |
| 1 | **9.67** | 11.45 | 14 | 8 |
| 2 | **23.025** | 24.024 | 18 | 8 |
| 3 | **168.59** | 172.355 | 22 | 8 |
| 4 | **408.532** | 408.542 | 25 | 8 |

this line can be found in [Arias et al., 2018, Lierler, 2023].

Knowledge compilation is a well-known technique in probabilistic (logical) settings, used to perform probabilistic inference [De Raedt et al., 2007, Dries et al., 2015] also in programs with both discrete and continuous random variables [Zuidberg Dos Martires et al., 2019], and to solve decision theoretic problems [Van den Broeck et al., 2010], allowing a compact representation of the program at hand.

# 6 CONCLUSION

In this paper, we discussed how to perform inference in probabilistic answer set programs with imprecise probabilities via optimization. Our pipeline consists of four steps: construction of a NNF representation of the query, extraction of two equations from the NNF, one for the lower and one for the upper probability, simplification of the equations, and solution of two constrained nonlinear optimization problems (minimization for the lower probability and maximization for the upper probability). Empirical results show that our approach is significantly faster than an already existing solver. Future works involve the study of related inference tasks, such MAP inference with imprecise probabilities.

**Acknowledgements**

This work has been partially supported by the Spoke 1 "FutureHPC & BigData" of the Italian Research Center on High-Performance Computing, Big Data and Quantum Computing (ICSC) funded by MUR Missione 4 - Next Generation EU (NGEU), and by TAILOR, a project funded by EU Horizon 2020 research and innovation programme under GA No. 952215. Both authors are members of the Gruppo Nazionale Calcolo Scientifico – Istituto Nazionale di Alta Matematica (GNCS-INdAM).

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
