# OpenReview forum: "Inference in Probabilistic Answer Set Programs with Imprecise Probabilities via Optimization"
_auai.org/UAI/2024/Conference — UAI 2024 oral_

### Official Review · Reviewer_fV7T · 2024-03-18

**Q2-1 Originality-Novelty:** 3
**Q2-2 Correctness-Technical Quality:** 3
**Q2-5 Clarity Of Writing:** 4

**Q10 Ethical Concerns:**

No.

**Q1 Summary And Contributions:**

The paper proposes a new way to compute inferences in probabilistic answer set programs with imprecise probabilities. Instead of enumerating extreme points as done previously, the paper proposes solving a non-linear optimization problem. Numerical experiments demonstrate the effectiveness of the approach, noting considerable computational improvements, due to not needing to enumerate all of the extreme points.

**Q2-3 Extent To Which Claims Are Supported By Evidence:**

4: Excellent: all claims are supported by very convincing evidence (in the form of comprehensive experimental evaluation, rigorous mathematical proofs, detailed (pseudo-)code, precise references, well-motivated and realistic assumptions) and the authors deliver what they promise.

**Q2-4 Reproducibility:**

3: Good: key resources (e.g. proofs, code, data) are available and key details (e.g. proofs, experimental setup) are sufficiently well-described for competent researchers to confidently reproduce the main results.

**Q3 Main Strengths:**

Excellent writing, clear introduction to the topic, contributions are clearly described and tested, easy to understand for outsiders to the field.

**Q4 Main Weakness:**

Contribution is more towards incremental rather than fundamental. It is well known that vertex enumeration is very inefficient (e.g. in the context of linear programming), so the results seem not so surprising. However, the numerical tests and application of non-linear solvers remains non-trivial as solving non-linear optimization problems can still be very challenging.

**Q5 Detailed Comments To The Authors:**

abstract: ... configurations *show* the effectiveness ...

p1, sec 1: check citation formatting of Sato and De Raedt in 1st paragraph

p1, sec 2, 1st sentence: remove "are"?

p1, sec 2, last sentence: world*s*

p2, 2nd col, paragraph above example 2: "contributes to both the lower and upper" -> "contributes to the lower" [the original is also true but I think this is clearer, also links more clearly with eq (4)]

p2, eqs (4) & (5): Do these bounds determine a belief function?

p2, eq (6): Notation is unclear. What is the meaning of \oplus and \otimes?

p3, left col, 1st paragraph: and leaf are -> and leaf*s* are

p4, left col, 1st paragraph: check citation formatting of Azzolini&Riguzzi

**Q9 Complying With Reviewing Instructions:**

Yes

---

> ### Author Rebuttal · Authors · 2024-04-05
>
> Thanks for the review.
>
> In the following we address the comments:
>
> [the numerical tests and application of non-linear solvers remains non-trivial as solving non-linear optimization problems can still be very challenging.]
> - Yes, complexity results for inference in PASP by Cozman and Maua show that the task complexity goes up several levels of the polynomial hierarchy. Furthermore, in “Specifying Credal Sets With Probabilistic Answer Set Programming” by Denis Deratani Mauá and Fabio Gagliardi Cozman, the author proved that “Deciding whether the unconditional lower probability of an atom is above a given threshold is NP^{PP}-complete in probabilistic answer set programs with parameterized annotated disjunctions.” An in-depth investigation about improvements in solving the optimization problem is an interesting future work.
>
> [p2, eqs (4) & (5): Do these bounds determine a belief function?]
> - The probability intervals can be related to DS theory belief functions. In particular, lower and upper probabilities can be seen as belief and plausibility functions. Further investigating the relations between these is an interesting future work.
>
> [p2, eq (6): Notation is unclear. What is the meaning of \oplus and \otimes?]
> - we will specify it in the paper. These are the components of two commutative semirings.
>
> We will fix all the reported typos.

---

### Official Review · Reviewer_Pg7y · 2024-03-19

**Q2-1 Originality-Novelty:** 3
**Q2-2 Correctness-Technical Quality:** 3
**Q2-5 Clarity Of Writing:** 4

**Q10 Ethical Concerns:**

No.

**Q1 Summary And Contributions:**

The paper addresses the topic of dealing with imprecise probability assessments in probabilistic answer set programming. Specifically, the paper considers credal probabilistic facts and credal annotated disjunctions, and addresses the inference problem by making a transformation into a constrained nonlinear optimization problem.

**Q2-3 Extent To Which Claims Are Supported By Evidence:**

4: Excellent: all claims are supported by very convincing evidence (in the form of comprehensive experimental evaluation, rigorous mathematical proofs, detailed (pseudo-)code, precise references, well-motivated and realistic assumptions) and the authors deliver what they promise.

**Q2-4 Reproducibility:**

3: Good: key resources (e.g. proofs, code, data) are available and key details (e.g. proofs, experimental setup) are sufficiently well-described for competent researchers to confidently reproduce the main results.

**Q3 Main Strengths:**

The use of imprecise probabilities within probabilistic answer set programming is interesting, and had only been considered so far by Mauá and Cozman. The present paper presents some novelties: the incorporation of credal annotated disjunctions as well as a more efficient approach to the inference problem.

The paper is well-written and clearly explained, and a good knowledge of the relevant literature is shown. The experimental part supports the conclusions.

**Q4 Main Weakness:**

Some elements are not too clear, partly because of the space limitations imposed by the conference. In particular, I did not fully gather if the inference is equivalent to considering all the precise probability models compatible with the credal sets and then taking the lower/upper envelopes of the inferences or if you are rather obtaining an outer approximation of this.

**Q5 Detailed Comments To The Authors:**

One issue that is not entirely clear from the point of imprecise probabilities is the crredal sets that are employed in the problem. As I understand it, the paper makes use of what are called probability intervals in the literature, that satisfy a property of 2-monotonicity but not necessarily k-monotonicity for values k>2. However, the credal set inference described in equations (4) and (5) is reminiscent of the lower and upper probabilities in D-S theory, that would be completely monotone and completely alternating, respectively. Hence, I was wondering if in your inference the resulting imprecise probability models are indeed belief/plausibility functions, and in that case if it would be possible to exploit some of the tools within D-S theory to simplify the algorithm further.

In particular, related to my question in Q4, I wonder too if the vertices of the credal sets could be exploited further in the computation of \underline{P}(q) and \overline{P}(q).

I also point below a few typos:
-Page 4, 'non-liner'.
-Page 9, 'YuliYa' -> Yuliya; 'Joasquin' -> 'Joaquin'
-Page 10: please give a link to the paper by Pacenza and Zangari.

**Q9 Complying With Reviewing Instructions:**

Yes

---

> ### Author Rebuttal · Authors · 2024-04-05
>
> Thanks for your review.
>
> In the following we address the comments:
>
> [relation with the DS theory]
> - The probability intervals we consider can be related to the belief functions described by DS theory. In particular, lower and upper probabilities can be seen as belief and plausibility functions. Further investigating whether we can employ DS theory tools here is an interesting future work.
>
> [I wonder too if the vertices of the credal sets could be exploited further in the computation of \underline{P}(q) and \overline{P}(q).]
> - yes, the vertices may be used to solve the inference task. These were adopted in dpasp introduced in “Specifying Credal Sets With Probabilistic Answer Set Programming” by Denis Deratani Mauá and Fabio Gagliardi Cozman, which describes an algorithm based on vertex enumeration. Here we adopt an alternative approach based on optimization. Further analyzing the structure of the induced credal sets to speed up inference may be an interesting future work.
>
> We will fix all the typos reported in the last paragraph of Q5.

---

### Official Review · Reviewer_HqJy · 2024-03-22

**Q2-1 Originality-Novelty:** 3
**Q2-2 Correctness-Technical Quality:** 3
**Q2-5 Clarity Of Writing:** 3

**Q1 Summary And Contributions:**

The paper proposes a probabilistic answer set programming setting with credal probabilistic facts and credal annotated disjunctions with inprecise probabilities, and reduces inference to constrained nonlinear optimization problem. An experimental evaluation is also conducted showing the effectiveness of the proposed approach.

I acknowledge to have read the authors rebuttal.

**Q2-3 Extent To Which Claims Are Supported By Evidence:**

3: Good: the main claims are supported by convincing evidence (in the form of adequate experimental evaluation, proofs, (pseudo-)code, references, assumptions).

**Q2-4 Reproducibility:**

3: Good: key resources (e.g. proofs, code, data) are available and key details (e.g. proofs, experimental setup) are sufficiently well-described for competent researchers to confidently reproduce the main results.

**Q3 Main Strengths:**

An interesting probabilistic answer set programming setting with imprecise probabilities. The bound are computed with non -linea optimisation.

**Q4 Main Weakness:**

- personally, I believe that the proposed evaluation datasets (of combinatorial nature), even if used here and there, are somewhat far from any realistic application. The authors should try to come up with more realistic and larger examples.
- the authors use SciPy as library for the optimisation part. From my experience with combining logics with optoimisations tools, the optimisation tool is typically the bottleneck, if one does not rely on highly optimised and "commercial" tools such as Gurobi or CPLEX. Using them in place of 'accademic' solutions,  makes a huge difference in practice in terms of size and execution time
- how do I run the tests/resoner?

**Q5 Detailed Comments To The Authors:**

The paper is is reasonably wll written and clear. Some innacurencies pop-up here and there

- in eq.1, be precise and define the case w=\emptyset
- in eq. 6, who is f (and provide examples immediately afterwards)
- provide explcit def. of f_lp and f_up
- eq. 8,9, you optimise twice. Did you try to see whether you may go for optimising min - f_lp(X) \cdot f_up(X) with some additional constraints such as f_lp(X) \leq f_up(X) ?
- in p.5, top left: seem to me be a  MIQP problem, not a generic non-liner optimisation problem..for which specialised solvers exists
- "..approach can be straightforwardly extended to support parameterized annotated disjunctions". in fact, you may attach any set of MIQP constraints to an ASP...
- biblio: clarify "Francesco Pacenza and Jessica Zangari. Extending answer set programming with rational numbers, 2023."

**Q9 Complying With Reviewing Instructions:**

Yes

---

> ### Author Rebuttal · Authors · 2024-04-05
>
> Thanks for your review.
>
> In the following we address the main weakness (Q4):
>
> [personally, I believe that the proposed evaluation datasets (of combinatorial nature), even if used here and there, are somewhat far from any realistic application. The authors should try to come up with more realistic and larger examples.]
> - The graph coloring example is often used in ASP as a standard benchmark. The smokers dataset is a well-known example in probabilistic logic/answer set programming. We designed the last two datasets to test the solver on a worst case combinatorial scenario. Developing further datasets describing more realistic scenarios is an interesting future work.
>
> [the authors use SciPy as library for the optimisation part. From my experience with combining logics with optoimisations tools, the optimisation tool is typically the bottleneck, if one does not rely on highly optimised and "commercial" tools such as Gurobi or CPLEX. Using them in place of 'accademic' solutions, makes a huge difference in practice in terms of size and execution time]
> - We used SciPy since it is free to use and also has a nice and easy interface. Gurobi and CPLEX are not free, so they may reduce the usability of our tool. The exploitation of specialized solvers  is an interesting future work.
>
> [how do I run the tests/resoner?]
> - We added a detailed description of it into the repository, that we will make available on github and will add a link to the paper.
>
> In the following we addres the Detailed Comments (Q5):
>
> [in eq.1, be precise and define the case w=\emptyset]
> - in this case only the second product will apply, since none of the a_i will be in w.
>
> [in eq. 6, who is f (and provide examples immediately afterwards)]
> - f is a transformation function that maps the value computed in the inner AMC to values for the outer AMC. We will specify it in the paper since its definition was missing, thanks.
>
> [provide explcit def. of f_lp and f_up]
> - we will add a general definition of these two functions, which is f_{lp/up}(X) = \sum_{w} \prod_{a_i \in w} \pi_i \prod_{a_j \not\in w} (1 - \pi_j). That is, they are composed of summations of products. However, their form is usually more compact.
>
> [eq. 8,9, you optimise twice. Did you try to see whether you may go for optimising min - f_lp(X) \cdot f_up(X) with some additional constraints such as f_lp(X) \leq f_up(X) ?]
> - we didn’t think about this, but it can certainly be material for further investigation. In the experimental part we run two optimizations, for both lower and upper probability. However, in practical cases, the user may be interested in maximizing only one of the two.
>
> [in p.5, top left: seem to me be a MIQP problem, not a generic non-liner optimisation problem..for which specialised solvers exists]
> - The task we consider does not fall into the class of mixed integer problems since we consider only continuous variables nor into the class of quadratic problems, since multiplications may involve more than two variables.
>
> [biblio: clarify "Francesco Pacenza and Jessica Zangari. Extending answer set programming with rational numbers, 2023."]
> - this is a paper that appeared only on arxiv and not yet published elsewhere. We will specify it in the paper and add a link.

---

### Official Review · Reviewer_8hiE · 2024-03-23

**Q2-1 Originality-Novelty:** 3
**Q2-2 Correctness-Technical Quality:** 3
**Q2-5 Clarity Of Writing:** 3

**Q1 Summary And Contributions:**

This paper introduces a new way to do inferences in probabilistic answer set programs, where the probability facts can be imprecise-probabilistic, in the sense that they can be interval-valued.
The way the authors introduce here is by rephrasing the problem as a constrained non-linear optimization problem.
This is an alternative way to the already existing method from 2023 based on vertex enumeration.

The authors derive their method in Section 3, while giving toy examples to guide the reader.
In Section 4 the authors perform some experiments which show that their new method is significantly faster than the extant method.

**Q2-3 Extent To Which Claims Are Supported By Evidence:**

3: Good: the main claims are supported by convincing evidence (in the form of adequate experimental evaluation, proofs, (pseudo-)code, references, assumptions).

**Q2-4 Reproducibility:**

3: Good: key resources (e.g. proofs, code, data) are available and key details (e.g. proofs, experimental setup) are sufficiently well-described for competent researchers to confidently reproduce the main results.

**Q3 Main Strengths:**

This paper deals with an important topic, as probabilistic answer set programs are gaining interest, and imprecise-probabilistic variants have an expressive advantage. This paper's main significance, to my mind, is that it shows that inference in such a setting can be done efficiently, too.

To my taste, the fact that the setting can deal with disjunctions of imprecise-probabilistic facts seem significant. I appreciate that the authors explain this setting.

The authors use a clear writing style in their paper. I appreciate the use of examples (although as the paper progresses, they become smaller, perhaps due to page limitations).

Also, the paper has an extensive section on experiments, and the authors made a wise choice in the type of experiments they report on. I like the fact that the code is made available.

**Q4 Main Weakness:**

Despite the clear writing style, I feel that some concepts are not sufficiently explained. I understand that this might be due to page limitations, but I wonder whether a more wise choice of topics in which the authors delve into details is not possible.

For instance, the last paragraph on Page 2 is not clear to me.
I don't understand what the $\oplus$ and $\otimes$ mean exactly, and I fail to see the significance for the rest of the paper.
Could this part be omitted, and later parts that seem more significant to the results be explained in some more detail? Of course, I may be mistaken about this; if so then I suggest that the authors try to better motivate the significance of this part.

Furthermore, the final parts of Section 3.1 (before Example 4) and of Section 3.2 deserve examples too, or at least some more information.

Another (be it minor) instance of the clarity of the explanation, is on Page 2, right column: What are "stable models"? This is not properly explained before.

**Q5 Detailed Comments To The Authors:**

Here are my detailed comments:

Page 1, right column, third paragraph: Don't the authors intend to say "imprecise probabilities" instead of "uncertain probabilities"?

Page 1, right column, final line: "world" should read "worlds".

Page 2, Example 1, line after the box: "for" should read "four".

Page 2, left column, line after Eq. (3): in the denominator of $\pi_i$, the $1-\pi_i$ should be $1-\pi_j$, and "$i>0$" should read "$i>1$".

Page 2, first line of right column: "0.3/(1-0.8)" should read "0.3/(1-0.2)".

Page 4, left column, first paragraph: "non-liner" should read "non-linear. A couple of lines later, "... if $\pi_1$" is associated to the credal fact ..." should read "... if $\pi_i$" is associated to the credal fact ...", and "the upper probability of $qr$" should be "the upper probability of $q$", I believe.

**Q9 Complying With Reviewing Instructions:**

Yes

---

> ### Author Rebuttal · Authors · 2024-04-05
>
> Thanks for your review.
> In the following we will answer to the main concerns (Q4):
> - [For instance, the last paragraph on Page 2 is not clear]
> This part briefly introduces 2AMC. We will add a detailed description of the task in the paper. The \oplus and \ominus are the components of a commutative semiring.
>
> - [the final parts of Section 3.1 (before Example 4) and of Section 3.2]
> We will try to add some examples if these will fit in the page limit.
>
> - [What are "stable models"]
> A stable model is a minimal model under set inclusion of the reduct of a program. The reduct of a program P w.r.t. an interpretation I is obtained by removing from P all the rules that have the body false in I. An interpretation is called model if it satisfies all the groundings of the rules of P. We will add these definitions in the paper.
>
> Thanks for reporting all the minor issues of Q5: we will fix all of them.

---

### Meta-Review · Area_Chair_4iDx · 2024-04-16

The paper addresses a relevant and difficult problem and provides a good solution. All 4 reviewers favoured acceptance.